# Induced responses contribute to rapid adaptation of *Spirodela polyrhiza* to herbivory by *Lymnaea stagnalis*

Antonino Malacrinò [1,2,5 ✉], Laura Böttner[2,3,5], Sara Nouere[2,4], Meret Huber[3,4], Martin Schäfer [2,4] & Shuqing Xu [2,4 ✉]

Herbivory-induced responses in plants are typical examples of phenotypic plasticity, and their evolution is thought to be driven by herbivory. However, direct evidence of the role of induced responses in plant adaptive evolution to herbivores is scarce. Here, we experimentally evolve populations of an aquatic plant (*Spirodela polyrhiza*, giant duckweed) and its native herbivore (*Lymnaea stagnalis*, freshwater snail), testing whether herbivory drives rapid adaptive evolution in plant populations using a combination of bioassays, pool-sequencing, metabolite analyses, and amplicon metagenomics. We show that snail herbivory drove rapid phenotypic changes, increased herbivory resistance, and altered genotype frequencies in the plant populations. Additional bioassays suggest that evolutionary changes of induced responses contributed to the rapid increase of plant resistance to herbivory. This study provides direct evidence that herbivory-induced responses in plants can be subjected to selection and have an adaptive role by increasing resistance to herbivores.

[1] Department of Agriculture, Università degli Studi Mediterranea di Reggio Calabria, Reggio Calabria, Italy. [2] Institute for Evolution and Biodiversity, University of Münster, Münster, Germany. [3] Institute for Plant Biology and Biotechnology, University of Münster, Münster, Germany. [4] Institute of Organismic and Molecular Evolution, Johannes Gutenberg University Mainz, Mainz, Germany. [5] These authors contributed equally: Antonino Malacrinò, Laura Böttner. ✉email: antonino.malacrino@unirc.it; shuqing.xu@uni-mainz.de

Phenotypic plasticity, the ability of a genotype to display different phenotypes in response to different environmental conditions, is a key feature that is important for the survival and evolution of organisms. Changes in the environment can expose cryptic genetic variations, generating new phenotypes on which selection can act, and promoting the success of phenotypes adaptive to the new conditions[1,2]. Phenotypic plasticity may be particularly important in plants because, as sessile organisms, they need to constantly adjust to the changing environment[3–7]. For example, in response to attacks from herbivores, plants often increase the biosynthesis of defensive metabolites that deter herbivores from feeding, reduce their performance, or attract herbivores' natural enemies (e.g., flavonoids and phenolic compounds)[8–12]. Although previous studies have shown that the evolution of defensive metabolites can be driven by herbivore selection, direct evidence demonstrating the role of induced responses in adaptive evolution to herbivores remains scarce, as most current studies have not disentangled the effects of evolution and induced responses. Indeed, considering that herbivory-induced responses are often confounded by other ecological interactions and trade-offs between growth and defenses[13,14], it remains unclear whether herbivory-induced responses can contribute to the rapid adaptive evolution of herbivores.

In addition to inducing changes in plant phenotypes, herbivory affects the composition and function of plant-associated microbial communities[15]. This is non-trivial, considering that plant microbiomes are, in turn, able to modulate the phenotype of their hosts[16,17], including traits related to defense against herbivores[18,19] that can directly or indirectly affect plant growth and resistance against herbivory[20–22]. Thus, phenotypic responses caused by herbivory-induced microbiome changes may also influence plant adaptation to herbivores[23,24]. However, direct experimental evidence to support this hypothesis is scarce. This is largely due to difficulties in assessing the effects of changes in the microbiota on plant resistance in its native environment, as microbiome communities are highly dynamic and sensitive to abiotic environments.

To address these challenges, we used the duckweed *Spirodela polyrhiza* and its natural herbivore *Lymnaea stagnalis* (a freshwater snail) as a model system. *Spirodela polyrhiza* predominately reproduces via clonal propagation, with an asexual generation time of ~3–5 days outdoors during the growing season. *Spirodela polyrhiza* has a small size, a small genome[25], and a very low genetic diversity and mutation rate[26], making it an excellent model for real-time tracking of evolutionary processes through outdoor multigenerational experimental evolution. By experimentally evolving plant populations to multigenerational herbivory outdoors, we tested whether herbivory drives rapid adaptive evolution in plant populations. We then performed both indoor and outdoor bioassays to disentangle the role of induced responses and changes in genotype frequencies in rapid adaptation to herbivory in *S. polyrhiza*.

## Results

### Snail herbivory drives rapid adaptation in *S. polyrhiza* outdoors.
We experimentally evolved replicated populations of a mixture of four genotypes of *S. polyrhiza* (Sp21, Sp56, Sp58, and Sp65; Table. S1) under multigenerational herbivory by the snail *L. stagnalis* or control conditions, respectively. These four genotypes were used since they represent the genetic variation typically found in natural populations in Europe[26]. We replicated experimental populations over ten ponds outdoors, each hosting two cages with fine nets separating the two treatments (with or without herbivores) while maintaining plants in a common water body (Fig. 1 and Fig. S1). While tracking changes in plant morphology over the course of two growing seasons (Fig. 1a, b), we performed a series of experiments to test whether multigenerational herbivory can drive rapid adaptive evolution in plant populations.

To test the effect of herbivory on changes in plant growth rate, we measured the growth rates of the mixed populations outdoors in the control and herbivory treatments. Populations of *S. polyrhiza* that had previously been exposed to herbivory grew slower in terms of number of fronds ($F = 36.72$; df $= 1$; $p < 0.0001$; Fig. S2a), biomass ($F = 972.74$; df $= 1$; $p < 0.0001$; Fig. S2b), and area ($F = 28132$; df $= 1$; $p < 0.0001$; Fig. S2c) compared to populations that did not experience herbivory. Cross-sections of representative fronds revealed that herbivory altered plant morphology (Fig. 1). Together, these results show that *S. polyrhiza* populations grew smaller, slower, and heavier under multigenerational snail herbivory (Fig. 1). Nutrient levels and pH were similar in both cages within each pond throughout the first season (Table S9, Fig. S6, Supplementary Methods), except for Cl and $SO_4$, which were more abundant in control cages. Although marginal differences were detected, the effect size was small (Table S9), and likely did not contribute to the effects observed in the plant populations.

We then tested whether the observed phenotypic changes were adaptive by performing outdoor resistance assays using evolved populations under control and herbivory conditions. We found that snails consumed fewer fronds ($F = 35.71$; df $= 1$; $p < 0.0001$; Fig. 2a), less biomass ($F = 277.36$; df $= 1$; $p < 0.0001$; Fig. 2b), and less surface area of fronds ($F = 633.54$; df $= 1$; $p < 0.0001$; Fig. 2c) from the populations that had previously experienced herbivory compared to the control populations. In addition, we found that herbivory altered the production of putative defensive plant metabolites during the two consecutive growing seasons (Fig. S3), including consistently increased production of tyramine. Taken together, these results suggest that multigenerational herbivory reduces growth rate, alters plant morphology and metabolism, and increases herbivore resistance in *S. polyrhiza* populations.

### Evolution of herbivory-induced responses contributed to the rapid adaptation.
We tested whether the observed rapid phenotypic changes were due to changes in genotype frequency or phenotypic plasticity by quantifying the frequencies of each genotype in the mixed populations outdoors, using a pool-seq approach on samples collected after 8 and 12 weeks of experimental evolution. We found that the genotype frequency was influenced by the treatment (Table S3). In comparison to the starting population (25% of each genotype), herbivory significantly altered the genotype frequencies of Sp21 and Sp65 in opposite directions (Fig. 3) but did not influence the frequency of genotypes Sp56 and Sp58. At week 12, in comparison to controls, herbivory reduced the frequency of genotype Sp21 (control $33.9 \pm 2.4\%$, herbivory $24.5 \pm 1.8\%$, $p = 0.0004$) and increased the frequency of genotype Sp65 (control $19.3 \pm 1.6\%$, herbivory $27.1 \pm 2.1\%$, $p = 0.0006$), suggesting that herbivory rapidly altered genotype frequencies in the duckweed populations outdoors.

We then tested whether the observed adaptive changes were caused by intrinsic differences in growth and resistance among genotypes. To this end, we grew each genotype under sterile indoor conditions using plants directly from the sterile collection and measured their growth rate and resistance to herbivory (see Fig. S4 for further detail). Growth rate differed among the genotypes ($F_{3, 35} = 3.51$, $p = 0.02$; Fig. S4a), although the post hoc contrasts did not show pairwise differences between them, suggesting that the overall differences were marginal. Herbivory resistance assays showed that the four genotypes differed in their intrinsic resistance ($F_{3, 44} = 58.62$, $p < 0.0001$; Fig. S4b), mainly

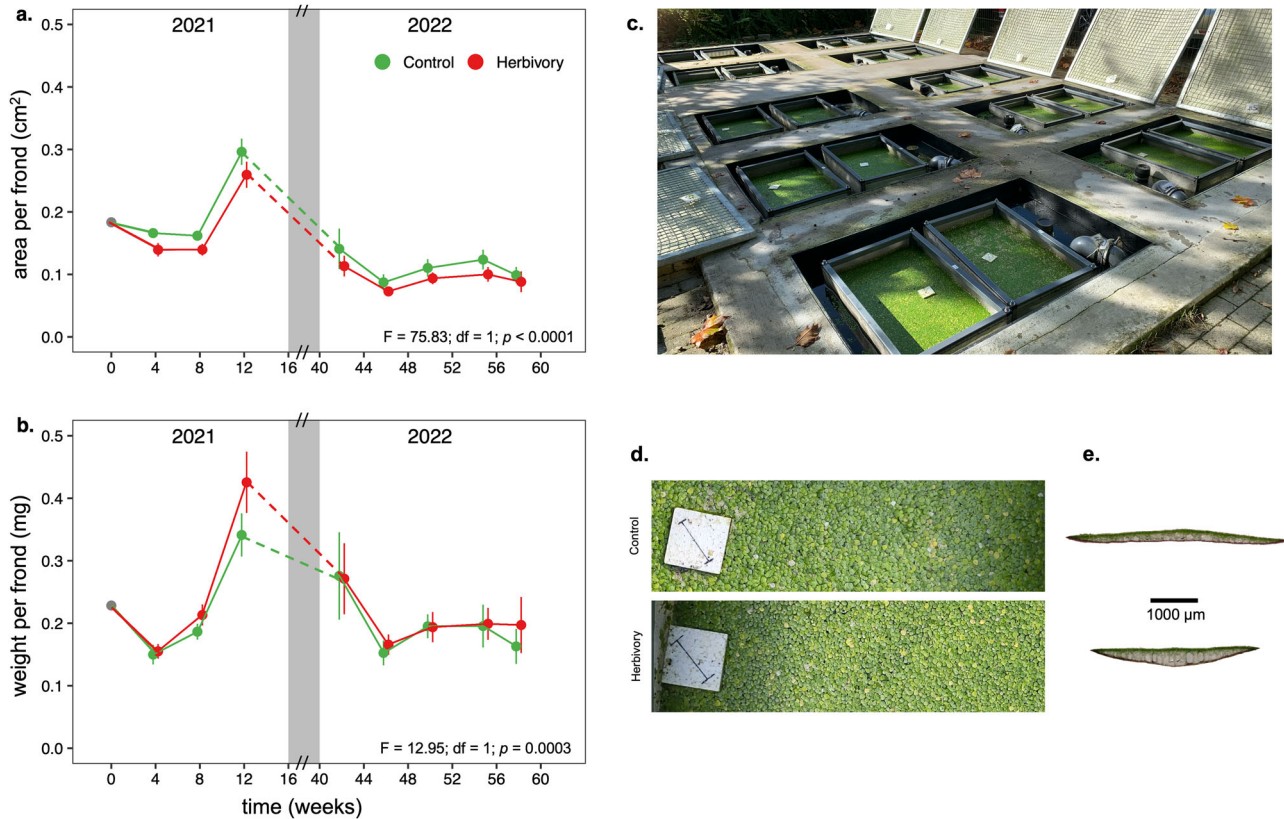

**Fig. 1 Rapid evolution of *Spirodela polyrhiza* outdoors.** Surface area (**a**) and dry weight (**b**) of 500 undamaged *S. polyrhiza* individuals of mixed-genotype populations exposed to herbivory across two growing seasons (2021 and 2022). Plant populations that experienced multigenerational herbivory outdoors showed a lower surface area and a higher dry weight per frond compared to the control group (Table S2). For each timepoint, dots represent mean values and bars show the standard deviation ($n = 10$). The gray area represents winter. Each plot reports the results from fitting a linear mixed-effects model using "treatment" (herbivory/control) as a fixed effect, and "time" and "pond" as random factors. Overview (**c**) of the experimental setup with the 10 ponds each hosting two cages (herbivory/control). Representative pictures of top pond view (**d**) and cross-sections of fronds (**e**) grown under control (**d**, **e** upper) and herbivory (**d**, **e** lower) conditions. Under herbivory fronds grew smaller in diameter but thicker. In panel (**d**) the styrofoam square measures 6 x 6 cm, and the line drawn in the middle measures 5 cm.

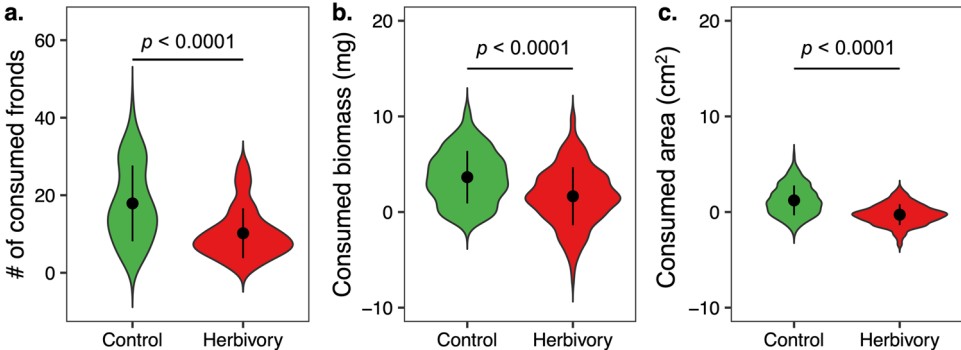

**Fig. 2 *Spirodela polyrhiza* populations evolved more resistance to snail herbivory.** Consumed number of fronds (**a**), biomass (**b**), and surface area (**c**) of *Spirodela polyrhiza* populations that evolved without (green) and with (red) herbivory outdoors. Values for plots (**b**, **c**) are calculated by permutation ($n = 999$) with a set of reference fronds collected before the bioassay. For each group, dots represent mean values and bars show the standard deviation.

due to the genotype Sp56, which is more susceptible to herbivory than the other genotypes ($p < 0.0001$). No differences were found in Sp21 and Sp65, the frequency of which was altered by herbivory in outdoor ponds, suggesting that neither the intrinsic growth rate nor resistance contributed to the rapid changes in resistance observed after outdoor experimental evolution.

We then investigated whether herbivory-induced plasticity per se can contribute to the observed phenotypic changes by experimentally subjecting each genotype separately to the snail's

attack or control conditions in indoor microcosms. After 8 weeks of growth, we found that the fronds of all genotypes grown under herbivory were smaller (Fig. S5 and Table S4), indicating that phenotypic plasticity could contribute to the observed morphological changes.

To further investigate the effects of phenotypic plasticity and changes in genotype frequency, we assembled four different artificial populations in our greenhouse ponds using a full factorial design and quantified the levels of resistance to

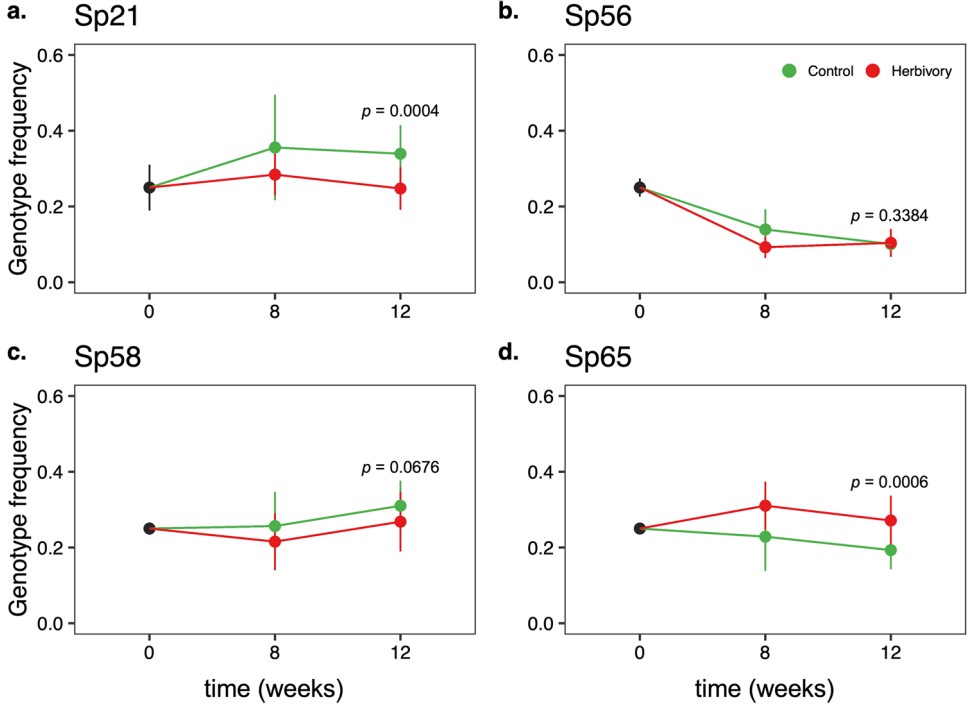

**Fig. 3 Snail herbivory altered genotype frequencies in the populations.** Relative frequency of *Spirodela polyrhiza* genotypes Sp21 (**a**), Sp56 (**b**), Sp58 (**c**), and Sp65 (**d**) at the beginning of the experiment ($n = 3$) and after experimental evolution (weeks 8 and 12) under multigenerational herbivory or in control conditions ($n = 10$). For each group, dots represent mean values and bars refer to the standard deviation.

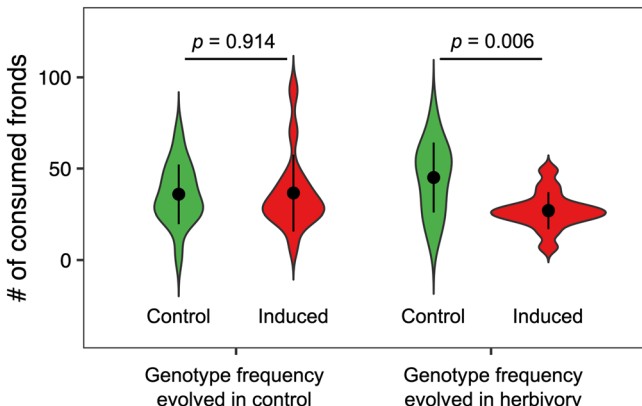

**Fig. 4 Evolution of induced responses contributed to increased resistance to herbivory.** The plot shows the number of consumed fronds after 24h of herbivory in synthetic populations. The genotype frequencies represent observed frequencies outdoors in the control (Sp21 36%; Sp56 10.7%; Sp58 32.9%, Sp65 20.4%) or herbivory (Sp21 27.8%; Sp56 11.7%; Sp58 30.1%, Sp65 30.4%) cages. Genotypes were either grown under herbivory (induced, red) or control (non-induced, green) conditions. For each group, dots represent mean values and bars show the standard deviation.

herbivory. We found clear interaction effects between induced responses and evolutionary changes in genotype composition (plasticity × evolution $F = 4.47$; df = 1; $p = 0.039$; Fig. 4). Although the changes in genotype frequency did not alter plant resistance in populations assembled using control plants ($p = 0.914$; Fig. 4), they increased resistance to herbivory when plants were previously induced by herbivory ($p = 0.006$; Fig. 4). Together, these results suggest that both herbivory-induced phenotypic plasticity and evolution contribute to the rapid increase in resistance of *S. polyrhiza* outdoors.

**Herbivory-induced microbiota changes contributed to the increased resistance to herbivory**. In addition to directly inducing changes in plant metabolism, herbivory can also induce changes in the plant microbiota, which might alter plant growth and resistance. Therefore, we characterized the duckweed-associated microbiota of mixed outdoor populations after 8 and 12 weeks of experimental evolution using the pool-seq data. We found that herbivory altered the structure of microbial communities associated with *S. polyrhiza* (Table S5).

We tested whether changes in microbiota might have caused changes in plant growth and resistance by growing individual genotypes within each treatment cage of our experimental setup outdoors. These plants were kept separate from direct herbivory using floating boxes with a fine net, so they were only exposed to the microbiota, either induced by herbivory or control, in the native environment of the experimental cages. After 8 weeks outdoors, we measured the growth rate and resistance to snail herbivory of these individual genotypes outdoors. We observed that only genotype Sp65 showed increased resistance to herbivory ($F = 3.66$; df = 1; $p = 0.05$; Fig. 5a), but not biomass or surface area (Table S6), when fronds were exposed to the indirect effects of herbivory, including the herbivory-induced changes in microbiota. No differences were observed among the other three genotypes ($p > 0.05$; Fig. 5a and Table S7). Because herbivory-induced microbiota had different effects on resistance levels in Sp21 and Sp65, we characterized the plant-associated microbial community of genotypes Sp21 (Fig. 5b) and Sp65 (Fig. 5c), in which genotype frequency was altered by herbivory, using an amplicon metagenomics approach. No differences between the two treatments were found when investigating the structure of the eukaryotic community associated with genotypes Sp21 ($F_{1, 12} = 0.73$; $p = 0.29$) and Sp65 ($F_{1, 12} = 0.52$; $p = 0.42$). However, herbivory induced little change in the bacterial microbiota associated with Sp21 ($F_{1, 12} = 1.40$; $p = 0.06$; Fig. 5b and Table S8), and significantly altered the bacterial microbiota of Sp65

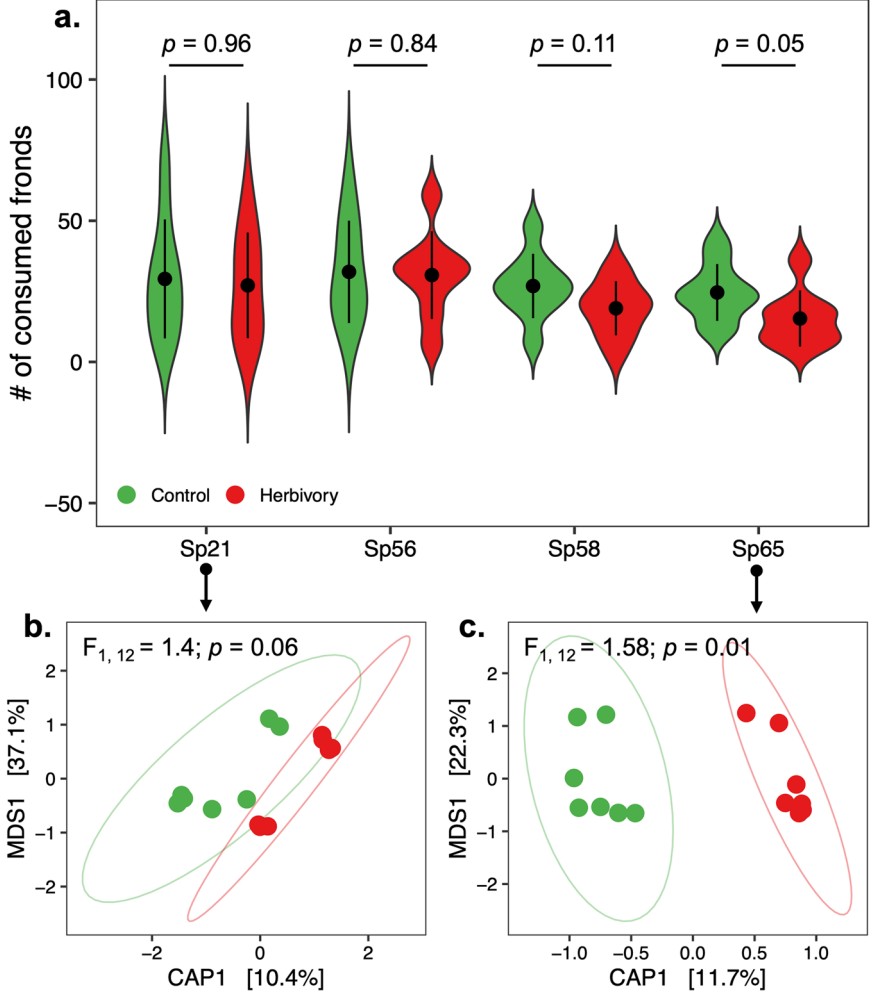

**Fig. 5 Genotype-specific effects of herbivory-induced changes in plant microbiota and resistance. a** Number of consumed fronds by snails for each *S. polyrhiza* genotype growing in microbial communities of control (green) and herbivory-induced (red) conditions, respectively. For each group, dots represent mean values and bars show the standard deviation (*n* = 10). **b, c** Canonical analysis of principal coordinates (CAP) analysis and PERMANOVA of bacterial communities (Unifrac distance matrix) associated with genotype Sp21 (**b**) and genotype Sp65 (**c**).

($F_{1, 12} = 1.58$; $p = 0.01$; Fig. 5c and Table S8), which was consistent with the observed differences in resistance between the two genotypes. These results suggest that herbivory-induced changes in plant microbiota might have contributed to the rapid changes in resistance to herbivory in *S. polyrhiza* outdoors.

## Discussion

Induced responses are widespread in nature and are thought to contribute to adaptation[1,2,4]. However, direct evidence of their effects on adaptive evolution remains scarce. Here, by carrying out experimental evolution outdoors, we demonstrated that induced responses contribute to the rapid adaptive evolution of *S. polyrhiza*. We found that snail feeding rapidly reduced the size and growth rate of the mixed *S. polyrhiza* populations. Although the four genotypes differed in size, the observed morphological changes in the populations were likely due to induced responses (phenotypic plasticity) but not the changes in genotype frequencies. This is because size differences were mostly found between Sp56 and the other three genotypes, whereas the frequency of Sp56 was not altered by herbivory. Furthermore, such morphological changes were not likely to be due to herbivory-induced microbiota, as individuals of the same genotype growing in the two different microbial communities (herbivore-induced and control) had similar frond sizes and growth rates. Instead, the

morphological changes were likely due to direct herbivory-induced phenotypic plasticity. When individual genotypes were attacked by snail herbivory indoors, similar morphological changes (smaller fronds) were observed in all four genotypes. Similarly, grazing herbivores can also induce phenotypic responses in other plant species[27–29], indicating a conserved response (either active or passive) in plants.

In addition to the observed morphological changes, herbivory also increased resistance to snails in *S. polyrhiza* populations. Through pool-seq analysis and bioassays using synthetic populations, we found that increased resistance to herbivory was likely caused by both induced responses (phenotypic plasticity) and changes in genotype frequencies (evolution), suggesting that herbivore-imposed selection acted on herbivory-induced traits in *S. polyrhiza*, supporting the existence of conditional neutrality for these traits[30]. Therefore, our results support the importance of phenotypic plasticity in unveiling cryptic genetic variation—a trait unveiled under specific conditions—in the rapid adaptive evolution to environmental factors[31].

Using a targeted metabolomics approach, we observed various changes in primary and secondary metabolites. Among them were several putative defensive metabolites such as luteolin, coumaric acid, cyanidin glycosides, shikimic acid, sucrose, and apigenin[32–37]. However, the changes varied significantly between

the two analyzed time points. Most metabolites can be affected by a variety of biotic and abiotic factors, and during the timeframe of the experiment, plants experienced a highly dynamic environment. Additionally, differences between short- and long-term responses to herbivory, differences in damage level, and changes in genotype frequency might causally contribute to the observed dynamics. A change that was observed under herbivory across both growing seasons was an increased abundance of tyramine, a response also observed in other plant-herbivore systems[38]. Tyramine has been reported in other plants to have the potential to directly affect plant-animal interactions[39], to be a precursor for herbivory-inducible secondary metabolites[40], and to be incorporated into cell walls, where it can influence cell wall digestibility[41]. However, for our plant-herbivore system, information on metabolic responses to herbivory and their effects on herbivores is scarce, and further experimental support is needed to draw clear conclusions. Additionally, it is also possible that the increased resistance might be due to thickened cell walls or other unknown mechanisms that alter the feeding choice of herbivores.

Our results also showed that herbivory altered the structure of the plant microbiome, both within the mixed population and in one of the genotypes used. While the effect of herbivory on the plant microbiome has been previously reported[20,21,42–44], we found that herbivory altered the microbiome of only one of the two assessed genotypes, genotype Sp65, which in the outdoors experimental evolution ponds increased in frequency under the herbivory group. This suggests that herbivory-induced changes in the microbiota are genotype-specific and that the outcome might influence phenotypic plasticity. For instance, a previous study focusing on another duckweed species (*Lemna minor*) found that plant-associated microbial communities can alter plant phenotypes, and the magnitude of this effect varies across genotypes[45]. Although several microbial taxa were specifically altered by herbivory in Sp65, it remains unclear whether particular bacterial taxa caused the observed changes in resistance.

One caveat of this study is that we included only four genotypes to represent the relatively low genetic variations found in natural populations of *S. polyrhiza*[26], which might limit the ability to generalize our findings to other systems that harbor more genetic variations. In contrast, populations with low genetic variation, such as many invasive species[46–49], might still rapidly adapt to the new environment through the plasticity of phenotypic traits[50]. The extent to which our findings apply to populations with higher genetic variation remains to be tested in the future. It is also important to acknowledge that the model species we selected reproduces solely asexually, and while *S. polyrhiza* allowed us to provide strong evidence that induced responses can be subjected to selection in plant populations, this might limit the generalization of our results to a wider proportion of plant species that reproduce sexually.

Taken together, this study provides direct evidence supporting the long-standing hypothesis that herbivory-induced responses are under selection and contribute to rapid adaptive evolution in plant populations. We were able to observe the adaptive role of induced responses in a relatively short timeframe owing to the short generation time of the study species. Although we believe that our findings are also applicable to other organisms that have longer life cycles, future long-term outdoor evolution experiments are required to provide further experimental support.

## Methods

**Outdoor experimental evolution**. We used an outdoor experimental evolution approach to study the herbivory-driven evolution of plants. The outdoor experiment was established in 2021 in Münster (Germany) and included 10 experimental ponds. We used four *S. polyrhiza* genotypes (Sp21, Sp56, Sp58, and Sp65; Table S1) to assemble the experimental populations, as they represent the genetic variation typically found in natural populations in Europe[26].

Before setting up the outdoor experiments, we separately grew the four genotypes of *S. polyrhiza* under controlled conditions (26 °C, 16 h:8 h, 135 μmol/m light) for approximately 4 weeks in full N-Medium (KH$_2$PO$_4$ 150 μM, Ca(NO$_3$)$_2$ 1 mM, KNO$_3$ 8 mM, H$_3$BO$_3$ 5 μM, MnCl$_2$ 13 μM, Na$_2$MoO$_4$ 0.4 μM, MgSO$_4$, 1 mM, FeNaEDTA 25 μM)[51]. We then moved all the plants to a semi-controlled greenhouse for approximately 1 week, helping them transition to outdoor conditions.

The outdoor mesocosm setup included ten ponds (1.5 (L) × 1.2 (W) × 1 (D) m; Fig. S1) with a water body volume of approximately 1500 L each. Seven weeks before the beginning of the experiment, the water body from the previous season (in which we had grown the same four genotypes without snails) was thoroughly cleaned of all floating organic residues using a scoop. We added 40 L of commercial potting soil (~100 mg/L each of organic N, P$_2$O$_5$, K$_2$O, ~160 mg/L Mg, pH (CaCl$_2$) = 4.2, salts 1.0 g/L; Floragard, Germany), and 550 mL of commercial liquid fertilizer (organic NPK 3.1-0,5-4.1; COMPO BIO, Compo GmbH, Germany) to each pond to provide sufficient nutrients for plant growth throughout the season. To prevent algal growth, we covered the ponds with white foil. A few days before beginning the experiment, we removed the covering foil and approximately 10% of the water body from each pond. The remaining water bodies were homogenized between all ponds, re-adding the removed 10% volume with tap water, thereby homogenizing the initial growing conditions between all replicates. Each pond was then divided into two compartments using two cages (105 (L) × 74 (W) × 44 (D) cm) with sides, and the bottom covered with a fine metal net (mesh 0.14 mm). Each pond was covered with a metal grid lined with a shading net, allowing for optimal growth conditions, and preventing large debris and small animals from falling inside the pond.

The experimental evolution of our plant populations was initiated on June 9th, 2021, by adding 500 fronds of each of the four *S. polyrhiza* genotypes to each cage, resulting in a mix of 2000 individuals per cage, referred to as mixed genotypes. Thus, each pond served as a paired unit to compare the effects of herbivory on the evolution of the aquatic plant populations. On the starting day of the experimental evolution, we collected 150 fronds of each genotype individually (in triplicate) for use as a reference to estimate the genotype frequencies (see below). We also measured the surface area and dry weight of 30 fronds of each genotype. In addition, we created artificially mixed populations using 500 fronds of each genotype, mixed them into a bucket, and sampled 400 fronds (in triplicate). This artificial mix (with three replicates) was later processed for pool-seq, and the data were used to normalize genotype frequencies to reduce sampling bias (see below).

We randomly assigned one cage to each pond as either a control (n = 10) or herbivory treatment (n = 10). Cages assigned to the herbivory treatment were inoculated with 12 snails (*L. stagnalis*, 18–22 mm shell length), and gradually added to groups of four over a period of 3 weeks to avoid the issue that adding them together would lead to the collapse of the entire plant population. Snail populations were allowed to propagate freely within herbivory cages for over two years. The entire experiment lasted 58 weeks (approximately 60 asexual generations of *S. polyrhiza*), including two summer seasons and one winter season. During the winter season, some of *S. polyrhiza* fronds remained on the water surface, and some of the fronds formed turions that sunk to the bottom (partly re-emerged during the following spring).

To test the indirect effects of herbivory on the resistance to herbivores and the microbiome of individual genotypes in situ, we added four floating boxes (11.5 (L) × 11.5 (W) × 7.5 (D) cm,

framed with Styrofoam, Fig. S1) to each cage on June 9th, 2021. Floating boxes had a fine metal net at the bottom to prevent herbivores from feeding, but allowed nutrients and phytoplankton communities to pass through. Each box received 150 fronds of one of the four genotypes, referred to as single genotypes.

To test whether herbivory influenced the morphology (surface area and dry weight), genotype frequency, microbiome, and metabolites of the *S. polyrhiza* mixed populations, we sampled fronds from cages after 4, 8, and 12 weeks of experimental evolution. Two sets of 500 undamaged fronds were collected from each cage. Each set was used to measure the surface area per frond and dry weight and was processed for (i) pool-seq or (ii) metabolite analysis (see below). We continued sampling in the second season (2022) from the mixed population at weeks 42, 46, 50, 55, and 58 to test the robustness of herbivory-associated changes in plant morphology. The samples collected from week 58 were also used for metabolite analyses. Pool-seq and bioassays (see below) were performed only for samples collected during the first experimental season (2021).

To measure the indirect effects of herbivory, we sampled fronds from single genotypes grown in floating boxes (without direct contact with herbivores), where we sampled 30 fronds per floating box at weeks 4 and 8 to measure surface area and dry weight. The surface area was measured for each sample by taking a picture of the fronds, including a floating reference square, immediately after sampling, and processed using the R package *pliman*[52] in R v.4.1.2[53]. The samples were rinsed in autoclaved tap water, frozen in liquid nitrogen, and stored at –20 °C. The dry weight was measured after freeze-drying the samples.

**Growth and herbivory resistance assays outdoors**. To measure the direct (mixed-genotype populations grown in cages) and indirect (single genotypes grown in floating boxes) effects of herbivory on the evolution of growth rate and resistance outdoors, we used a floating box system (see above).

First, to quantify the indirect effects of herbivory on the resistance of each individual genotype, we performed herbivory assays on week 8 for each individual genotype using the plants that had grown inside the floating boxes. To this end, we reduced the number of previously grown fronds to 100 in each floating box and added a single snail (from our laboratory population) that had been starved in the respective pond water for 24 h before starting the herbivore assay (total $n = 10$ per genotype and treatment, respectively). After 24 h of herbivory, all snails were removed, and all undamaged fronds were collected, counted, photographed to estimate the surface area, and freeze-dried to measure their dry weight.

Second, to quantify the effects of herbivory on the growth rate of the evolved mixed-genotype populations, we randomly selected fronds from five different spots within a cage in week 8, mixed them thoroughly, and added 50 fronds each to a floating box within the respective cage ($n = 4$ pseudo-replicates per $n = 10$ cages, total $n = 40$ per treatment). Fronds were allowed to grow in floating boxes (screened for herbivores) for 10 days. All fronds were harvested, counted, photographed to estimate the surface area, and freeze-dried to measure the dry weight.

Third, to quantify the effects of herbivory on the resistance of evolved mixed-genotype populations, we randomly selected fronds from five different spots within a cage at week 10 and randomly added 100 fronds to each of the four empty floating boxes. We introduced a single snail (from our laboratory population; individuals had been starved for 24 h in the respective pond water), allowing it to feed for 24 h ($n = 4$ pseudo-replicates per $n = 10$ cages, total $n = 40$ per treatment). After removing the snails from the floating boxes, we collected all undamaged fronds,

counted them, took a picture to estimate the surface area, and freeze-dried them to measure their dry weight. The entire procedure was repeated over two consecutive rounds, separated by 1 day to allow starvation of the snails (fully randomizing them between rounds), resulting in a total of $n = 80$ per treatment (herbivory vs. control).

**Pool-seq**. We used pool-seq on samples collected from a mixed population to quantify changes in genotype frequency and plant-associated microbial communities as a consequence of multi-generational herbivory. Samples included a group of 400 fronds randomly selected from the mixed populations in each cage at weeks 8 and 12 ($n = 20$ for each timepoint). In addition, we included three sets of samples and used them as a reference: (i) a set of three replicates for each plant genotype used to begin the evolution experiment ($n = 12$), (ii) a set of three replicates where we mixed an equal number of fronds of each genotype ($n = 3$), and (iii) a set of three replicates obtained by mixing 500 fronds of each genotype, mixing them thoroughly, and sampling 400 fronds ($n = 3$). These three sets of reference samples were used to reduce bias in the genotype frequency data (see below). DNA was extracted from ~10 mg of plant material from each sample using a phenol:chloroform protocol, and then shipped to Novogene Ltd. for shotgun metagenomics library preparation and sequencing on an Illumina NovaSeq 6000 S4 150PE flow cell.

Raw data were processed using TrimGalore v0.6.7 (https://github.com/FelixKrueger/TrimGalore) to remove Illumina adaptors and discard low-quality reads. We then used Bowtie2 v2.4.5[54], samtools v1.14[55] and the *S. polyrhiza* reference genome[25] to map the raw data and split reads for each sample to those matching the reference genome (to be used to estimate genotype frequency) and those not matching the reference genome (to be used to infer the structure of the plant microbiome).

**Genotype frequency from pool-seq**. We quantified the genotype frequencies using the reads mapped to *S. polyrhiza* reference genome[25]. For each sample, the BAM files of reads matching the *S. polyrhiza* reference genome were used to estimate the frequency of each genotype using the HAFpipe[56]. The VCF file with all the variable sites between the four genotypes was used[26] and the recombination rate was set to 0 (we also tried setting it to 0.00051 as in Ho et al.[57], and the results remained the same). We set the number of generations to 30 and the window size to 100 bp. Then, we used the data from the reference fronds for each individual genotype to identify loci that (i) identified a given genotype with a confidence >90%, (ii) were common to all four genotypes. This yielded 68 loci that could be used to estimate the frequency of the four genotypes in each sample. First, we tested whether this approach correctly identified our reference samples from individual genotypes, and we found that the misassignment error was always <1.9%, which was low. Then, we tested this pipeline on the samples where we mixed fronds from each genotype in equal numbers, and we found that the relative abundance of individual genotypes was different from 25% each. We reasoned that this was due to differences in size and biomass among the four genotypes (see results), which might have affected the amount of DNA extracted from each genotype. Given that the sampling itself might have added an error in the frequency estimates, we used reference samples (where we mixed an equal number of fronds from each genotype and then mixed them before subsampling) to calibrate the genotype frequencies for the experimental samples.

**Metagenomics analyses from pool-seq**. To characterize the composition of plant-associated microbial communities, we used samtools to extract from the BAM files the data that did not

match the reference genome of *S. polyrhiza*, and convert it into FASTQ files, which were then used as inputs for Kraken2 v2.1.2[58]. This analysis was performed twice, first using the Kraken2 reference database for archaea, bacteria, and fungi, and then using a custom database for algae built using all RefSeq genomes from NCBI for this group (Accessed on March 1st, 2022). The Kraken2 output was used to estimate the abundance of each taxon using Bracken v2.7[59]. Data were then collated into an abundance matrix, all singletons were discarded, and data were normalized using *DESeq2*[60].

**Metabolite analysis**. To measure the herbivory-mediated changes in specific plant primary and specialized metabolites, we analyzed the mixed population (a set of 400 fronds) randomly picked from each cage at weeks 12 (season 1, contextually to the sampling for pool-seq described above) and 58 (season 2) ($n = 20$ cages per sampling timepoint). Detailed information on the extraction procedure, standards used, and instrument settings is provided in the Supplementary Methods and Supplementary Tables S10–23. Briefly, samples were extracted with acidified methanol, and highly abundant compounds such as amino acids, sugars, and flavonoids were either directly measured in the pure extracts or in their dilutions. Low-abundance compounds, such as phytohormones, were analyzed after purification and concentration of the extracts using solid-phase extraction. Starch was analyzed from the sample pellets after enzymatic degradation to glucose. The analysis of highly abundant secondary metabolites was performed on a Shimadzu Nexera XR LC-System equipped with a Nucleodur Sphinx RP column and PDA detector. All other compounds were analyzed using the LC-MS/MS system that consisted of a Shimadzu Nexera X3 LC System (equipped with an apHera NH2 column for sugar analysis or a ZORBAX RRHD Eclipse XDB-C18 column for all other compounds), and a Shimadzu LCMS-8060 triple quadrupole mass spectrometer. Quantification was performed based on various individual external and internal standards (see Supplementary Materials).

**Amplicon metagenomics**. We further tested whether the genotypes Sp21 and Sp65 were associated with a different microbial community when grown outdoors under the indirect effects (single genotypes grown in floating boxes) of herbivory or control conditions. We selected these two genotypes because their frequency in the mixed population was significantly influenced by herbivory (see Results), and we focused on them to test whether they were associated with different microbial communities in the presence or absence of herbivores. We selected a group of 30 fronds from each genotype from each cage ($n = 40$) sampled at week 8 and extracted DNA as indicated above. The samples were then used for amplicon metagenomic library preparation and sequencing on an Illumina NovaSeq 6000 SP 250PE flow cell (Novogene Ltd.). Libraries targeting the bacterial community were built by amplifying the V4 region of 16S rRNA (primers 515 F and 806 R), whereas those targeting the eukaryotic community (including algae and fungi) were built to amplify the V4 region of 18 S rRNA (primers 528F and 706R).

Raw data were processed using TrimGalore v0.6.7 to remove Illumina adaptors and discard low-quality reads. Paired-end reads were processed using the DADA2 v1.22[61] pipeline implemented in R v4.1.2 to remove low-quality data, identify ASVs, and remove chimeras. Taxonomy was assigned using the SILVA v138 database[62] for 16S rRNA data or the PR2 v4.14 database[63] for 18S rRNA data.

**Growth and resistance assays in indoors using synthetic genotype frequency communities of evolved outdoor populations**.

To disentangle the effects of phenotypic plasticity and changes in genotype frequency on the observed resistance changes outdoors, we performed additional experiments under greenhouse conditions using microcosms similar to our outdoor setup while maintaining the temperature at 23 °C throughout the experiment. Here, for each individual genotype, we grew the fronds with or without snail herbivory for ~30 generations (similar to the single-genotype setup outdoors but with direct contact with the snail). Then, we assembled synthetic plant communities according to the evolved genotype frequencies observed outdoors at week 12 and ran both growth and herbivory assays.

We used five experimental ponds (60 (L) × 40 (W) × 27 (D) cm) filled with homogenized water bodies from our ten outdoor ponds. Each pond contained eight floating boxes (11 (L) × 11 (W) × 8 (D) cm). Each of the four genotypes was grown individually in two floating ponds within each pond. For each genotype and pond, one box with 600 fronds was exposed to herbivory by a single snail (*L. stagnalis*) placed inside the floating box, whereas the other box with 600 fronds was used as the control (paired design, $n = 5$ per treatment and genotype). Again, floating boxes were modified to allow water (and microbes) to flow freely between the boxes and the pond without allowing snails to leave the box.

After 8 weeks of indoor growth, we performed a growth assay in the same ponds using floating boxes. Here, the plant population within each floating box was reduced to 200, and herbivores were removed. After 14 days of growth, we counted all fronds and collected a sample of 50 fronds from each box to measure the surface area and dry weight.

Immediately after the growth assay, we performed a herbivory assay within the floating boxes using four synthetic populations: two assembled using the genotype frequencies observed in outdoor ponds of mixed populations under herbivory in week 12 (Sp21 27.8%; Sp56 11.7%; Sp58 30.1%, Sp65 30.4%; one population using greenhouse plants that experienced multigenerational herbivory and one population with greenhouse plants grown under control conditions), and two assembled using the genotype frequencies observed outdoors in mixed populations under control conditions at week 12 (Sp21 36%; Sp56 10.7%; Sp58 32.9%, Sp65 20.4%; one population using plants that experienced multigenerational herbivory and one population with plants grown under control conditions). The herbivory assays were replicated over the five ponds, exposing 100 fronds from each synthetic population to a single snail for 24 h (laboratory population, starved for 24 h prior). Then, we counted the residual fronds and measured the surface area and dry weight, as described above.

**Statistics and reproducibility**. All source data used for statistical analyses and to produce the graphs and tables have been deposited on Zenodo[64]. Data were analyzed using R v.4.1.2[53] with the packages *lme4* v1.1.29[65] and *car* v3.0.13[66], using different strategies as specified for each test below. Pairwise contrasts were performed using *emmeans* v1.7.4[67], correcting *p*-values using the false-discovery rate (FDR) method.

The overall effect of herbivory on the mixed-population outdoors over 2021 and 2022 was tested by fitting two linear mixed-effect models, one using frond surface area and the second using dry weight as the variable and specifying *treatment* (herbivory/control) as a fixed factor, and *pond* and *timepoint* as random factors ($n = 10$ for each treatment at any given timepoint), using the formula: ~treatment * (1| timepoint) * (1| pond).

The indirect effects of herbivory on the resistance of each individual genotype were estimated as the number of consumed

fronds, and the effect of herbivory on resistance was tested by fitting a linear mixed-effects model specifying *treatment* (herbivory/control) and *genotype* as fixed factors and *pond* as a random effect ($n = 10$ per genotype and treatment), using the formula ~treatment × genotype × (1|pond).

The effect of herbivory on the growth rate of the evolved mixed-genotype populations (relative growth rate and relative changes in biomass or surface area) was tested by fitting a linear mixed-effects model specifying *treatment* (herbivory/control) as a fixed factor and *pond* as a random effect ($n = 4$ pseudo-replicates per $n = 10$ cages, total $n = 40$ per treatment), using the formula: ~treatment * (1|pond). In this case, the relative changes in frond biomass and area were estimated by permutation ($n = 999$) using data from mixed-population monitoring (see above) before the beginning of this bioassay. The effect of multigenerational herbivory on resistance of evolved mixed-genotype populations was tested by fitting a linear mixed-effects model specifying *treatment* (herbivory/control) as a fixed factor and *pond* and *bioassay round* as random effects ($n = 4$ pseudo-replicates per $n = 10$ cages, total $n = 40$ per treatment at each round), using the formula ~treatment * (1|bioassay_round) * (1|pond).

We tested differences in the frequency of each genotype using data from pool-seq from experimentally evolved populations at weeks 8 and 12 ($n = 10$ for each treatment at each timepoint). After data processing (see above), we fitted a linear mixed-effects model (~treatment * genotype * (1|timepoint) * (1|pond)). Using the data from pool-seq we also characterized the structure of the microbial communities associated with plants, and we tested the effect of treatment and sampling time on the structure of microbial communities using PERMANOVA on a Bray-Curtis distance matrix between samples (999 permutations, stratified at the "pond" level). Taxa that were differentially abundant between treatments (herbivory/control) were identified at each timepoint (weeks 8 and 12) using *DESeq2* v1.34[60] with an FDR-corrected $p < 0.05$.

We tested the effect of multigenerational herbivory (herbivory/control) on each metabolite separately for weeks 12 (season 1, $n = 10$ for each treatment) and 58 (season 2, $n = 10$ for each treatment) by fitting a linear mixed-effects model for each metabolite, using the formula ~treatment * (1|pond).

Amplicon metagenomics data were then analyzed using R v4.1.2, using the packages *phyloseq* v1.38[68], *vegan* v2.6[69], *DESeq2* v1.34[60], and *lme4* v1.1.29[65]. Distances between pairs of samples in terms of community composition were calculated using an unweighted UniFrac matrix and visualized using a canonical analysis of principal coordinates (CAP) procedure. Differences between sample groups were inferred through permutational multivariate analysis of variance (PERMANOVA, 999 permutations) separately for each genotype, specifying treatment (herbivory/control) as a fixed factor, and using the factor "pond" to stratify permutations. Taxa that were differentially abundant between treatments (herbivory/control) were identified for each genotype using *DESeq2* with an FDR-corrected $p < 0.05$.

The effect of multigenerational herbivory on resistance using synthetic populations of *S. polyrhiza* indoors was tested by fitting a linear mixed-effects model specifying *population* and *treatment* (herbivory/control) as fixed factors, and *pond* as a random effect ($n = 5$ for each population and treatment), using the formula: ~population * treatment * (1|pond).

**Reporting summary**. Further information on research design is available in the Nature Portfolio Reporting Summary linked to this article.

## Data availability
Raw sequencing data are available on the NCBI SRA under the Bioprojects PRJNA849359 (pool-seq), PRJNA893536 (16S amplicon metagenomics), and PRJNA893537 (18S amplicon metagenomics). Data from bioassays are available on *Zenodo*[64] and at: https://github.com/amalacrino/malacrino_et_al_rapid_plant_adaptation_to_herbivory.

## Code availability
The code to replicate analyses is available on *Zenodo*[64] and at: https://github.com/amalacrino/malacrino_et_al_rapid_plant_adaptation_to_herbivory

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

## Acknowledgements

This project was supported by the German Research Foundation (project numbers 438887884 to S.X. and 422213951 to M.H.) and the Swiss National Science Foundation (P400PB_186770 to M.H.), which were inspired by discussions with the members of the CRC TRR 212 (NC3), Project number 316099922, and Research Training Group 2526 (GenEvo), Project number 407023052. The LC-MS instrument was funded by the German Research Foundation (project number 435681637 to S.X.). We thank the experimental assistance from Maximilian Schiefer, Zahra Kouzbour, Fabio Dudenhausen, Marie Sárazová, Florian Teutemacher, Florian Groß, Tom Lieth, as well as the help from Holger Schön and David Martín Fernandez for setting up the outdoor experiments. We also thank Sergio Ramos, Ian T. Baldwin, and Philipp M. Schlüter for providing constructive feedback on our manuscript.

## Author contributions

Conceptualization: S.X.; Methodology: S.X., A.M., L.B., S.N., M.S.; Investigation: A.M., L.B., S.N., M.S.; Visualization: A.M., L.B., S.X.; Funding acquisition: S.X., M.H.; Project administration: S.X.; Supervision: S.X., AM; Writing—original draft: A.M., L.B., S.X.; Writing—review, and editing: all co-authors.

## Funding

## Competing interests

The authors declare no competing interests.
