## [Peer Review File · Communications Biology]

Reviewers' comments:

Reviewer #1 (Remarks to the Author):

This manuscript presents potentially interesting work investigating the evolution of herbivory-induced changes in plant morphology, growth, and microbial communities. The authors performed an impressive array of experiments and targeted bioassays. However, the methods and results were generally difficult to follow, which made assessment of novelty and interpretation of results confusing. Both sections (methods and results) need major revisions that focus on developing a clear narrative structure to describe the experiments performed and their purpose, which then needs to be reflected in the structure of the results (i.e. structure of the results should clearly link to methods, making it easier for readers to understand). For example, the results section starts off with what appears to be the additional growth bioassays performed (starting week 8?) rather than data collected over both seasons 1 & 2 in the outdoor experiment, which is confusing. It is helpful if results are presented in the same order in which they are described in the methods. In addition, there was misalignment between results reported and statistical methods, including many missing details (e.g. Section "Snail herbivory drives rapid adaptation in *S. polyrhiza* outdoors"). Clear connections to the research aims and questions presented in the introduction are also generally lacking.

Below are my specific suggestions for improving the methods and results:

Methods:

- Lines 91-96: Description of measurement of indirect effects of herbivory using floating boxes (snails excluded) was confusing – why was this included in the design and why for weeks 4 & 8 only? These data don't seem to be relevant – the authors need to justify the purpose and how this approach addresses the broader research aims presented in the introduction.

- Subsection starting on Line 135 – "Growth and herbivory assays outdoors". Were growth bioassays performed in absence of herbivory? It is confusing that herbivory assays were performed "right before" growth assays, does this mean plants were exposed to herbivory before growth assays? Clarification is needed.

- Growth rates do not appear to be measured correctly – authors indicate relative changes of surface area and biomass were estimated but it unclear how that relates to growth rate (e.g. lines 144-146). Changes in biomass and surface areas from beginning and end of 10 day bioassay should be used to estimate growth, or simply analyze measures taken at day 10. Clarification is needed.

- Resistance was measured as # fronds consumed – unclear if this is the entire frond or if partial consumption of leaf tissue was measured (which could affect surface area measures). It would be helpful to explain patterns of snail herbivory, do they consume portions of fronds or entire fronds? This information matters for interpretation of differences in surface area (i.e. won't plants exposed to direct herbivory have reduced surface area?).

- Metabolomics section (line 228-253) lacked critical information on methods used – what type of metabolomics was performed, where, on what machine? What quality control measures were used to filter data? How was the concentration or abundance of metabolites estimated? All of these details need to be included.

- Unclear why microbial metabarcoding was performed on just 2 genotypes (section starting line 237) – please explain or justify. It is also generally unclear why shotgun sequencing was used on mixed populations only, and then additional metabarcoding was performed on a subset of 2 genotypes. Authors need to provide explanation and purpose of these measurements, what questions are being addressed?

- It is unclear what novel information the indoor bioassays provided, especially because populations were artificially created to replicate genotype frequencies observed at week 12 in the outdoor experiments. The authors need to provide additional details clarifying the purpose of these experiments/questions addressed.

Results:

- Section "Snail herbivory drives rapid adaptation in *S. polyrhiza* outdoors" – this section was generally missing details from statistical analyses described in the methods, such as linear mixed models and metabolomics. For example, chi-squared values were reported from analyses that were not described in the methods – results should align with what is described in methods. Additionally, the metabolomics results consisted of a single sentence with no statistics included (lines 324-326) – the authors need to expand on these results, what other metabolites were different?

- Figure 1 A & B was confusing - are these results from additional bioassays (described starting line 135) or data collected across seasons (lines 112-123)? Please clarify – if these data are not from the additional bioassays in the absence of herbivory pressure, then they potentially do not support claims of rapid evolution (i.e. biomass expected to be less from fronds directly exposed to herbivory). Also, chi-squared values are reported which do not align with linear mixed models described in methods. Please provide results for fixed and random effects of the linear mixed models.

- Lines 371-373 – where is this indoor experiment described in the methods and how is it different from that described on line 356-368? Authors needs to clarify, and again the methods section should also provide explanation of purpose of indoor experiments that then clearly align with results.

Microbiome data was not well integrated into the manuscript overall - not only were there missing details but the authors make incorrect claims that herbivory-induced microbiota changes contributed to increased resistance to herbivory. Below are my specific suggestions:

- Details on differences in microbial communities and results from statistical analyses need to be included (e.g. expand on sentence on lines 393-94). What microbial taxa were identified and which are differentially abundant between control plants and those exposed to herbivory? What are the results from PERMANOVA & DeSeq? The authors should include a more complete analysis of microbial diversity and composition, including both alpha (e.g. species richness) and beta (e.g. Bray Curtis, Unifrac) diversity measures.

- Lines 395- 413 – based on the experiments described it is not possible to draw conclusions about the effect of the plant microbiome on resistance to herbivory because proper controls (i.e. sterile water/reduced microbiota treatment) were not included. The authors use language that suggest causation (i.e. changes in microbes caused changes in resistance) but experiments only show associations. Fig 5 shows herbivory induced changes in the plant microbiome, but this overall effect does not seem to vary substantially between genotypes expressing different levels of resistance (marginally significant effect of herbivory in Fig 5b). The authors provide no evidence to support claims that microbiota are involved in resistance to herbivory, to do that follow-up experiments under sterile (no microbiome) conditions or transfer of microbiomes between genotypes are needed. This section needs to be substantially edited (i) to remove language suggesting a causal relationship between herbivory-induced changes in the plant microbiome and resistance and (ii) to include additional details on what herbivory-induced microbial differences were observed (e.g. results from DeSeq, additional analysis for both alpha (e.g. species richness) and beta (e.g. Bray Curtis, Unifrac) diversity measures).

Reviewer #2 (Remarks to the Author):

General comments

I understand the overall objective of this study but I do not really see how the experimental approaches chosen allow to meet the specific objectives. For instance, what is the purpose of the indoor system in comparison to the outdoor system? In addition, the results are not sufficiently described and discussed.

Other Comments on the system description:

To my point of view, it seems that elements required to fully describe the experimental system are missing.

What is the composition of the Com BIO liquid fertilizer?

It's not clear to me what is the volume of liquid inside the ponds? I understand there is 40 L of soil and 0.55 L of liquid fertilizer but nothing else?

How floating boxes float on without water?

Is the fronds density the same between the floating boxes and the rest of the cage at the beginning of the experiment and over time?

Comments on the microbial community structure analysis:

This analysis was carried out on a very limited number of time samples: after 8 and 12 weeks (metagenomes) and only 8 weeks (amplicon metagenomics). Why 2 types of analyses/methods were used? Why the amplicon metagenomics analyses was only shown for 2 genotypes, Sp21 and Sp65? Why the microbial community structure was not analysed earlier like at time 0 and also during year 2 at the end of the experimental evolution in order to have a better description and understanding of population dynamics of the associated microbial communities during the whole experimental evolution, and in response to herbivory.

In the introduction section (line 58), the authors indicate that microbiome community are highly dynamic and sensitive to abiotic environment. It's true but the authors only performed microbiome analysis on a reduced number of sampling times and of conditions. Why do they choose these sampling times and those conditions?

Authors show that there are changes in the frequency of *Spirodela polyrhiza* populations in response to herbivory. These changes may lead to direct changes in microbial communities as well as phenotypic changes in *S. polyrhiza* in response to herbivory. How can the authors differentiate the effect of these 2 confounding factors on the structure of the microbial communities? It would have been interesting to compare the structure of microbial community of each genotype (not only on Sp21 and Sp65) and of mix populations at different times in presence and absence of herbivory pressure.

Comments on Metabolomics analysis:

To my point of view, it is necessary to describe the experimental procedure used for the metabolomics analysis inside materials and methods section of the paper instead of the supplementary materials.

In addition, the results from Fig S3 are not sufficiently described and discussed. What does this study of the evolution of metabolomes after 12 and 58 weeks show regarding plant responses to herbivory? It appears that after 12 weeks more defensive metabolites are identified as overproduced in the conditions with herbivory vs without than after 58 weeks? Tyramine is the sole compound that is maintained increased after 12 and 58 weeks ! What does this mean? Could you discuss these results in more depth in the discussion section ? Are the metabolome changes correlated to the evolution of *Spirodela polyrhiza* populations inside the mix throughout the experimental evolution?

In the introduction (lines 41 to 43), authors indicate that plants often respond to herbivores by increasing the biosynthesis of defensive metabolites. What are these defensive metabolites? Could you detail the main compound family of these defensive metabolites for aquatic plants like *Spirodela polyrhiza*? Are these compounds identified in the metabolome analysis performed in the presented work?

In the discussion section (lines 445 to 446), authors indicate that 'using a targeted metabolomics approach, we observed changes in several candidate defensive metabolites, which were induced by herbivory across two growing seasons.' What are those compounds?

They also indicate that 'their results support the importance of cryptic genetic variation in contributing

rapid adaptive evolution to environmental changes' but why do the authors introduce this notion of 'cryptic genetic variation' there? Against what does this mean? Could they discuss further this point.

Other comments

Lines 303-304: authors indicate that 'Nutrient levels and pH were similar in both cages ... suggesting that the effects we observed were not driven by differences in resources'. Overall, the NMDS analysis (FigS6) shows no difference. However, in Table S9 it seems that the level of Cl and SO₄ in water were changed in response to herbivory. Could you add a comment on these changes in the result and discussion sessions.

Reviewer #3 (Remarks to the Author):

This study investigated the role of evolution and phenotypic plasticity in driving rapid changes in plant resistance to herbivory. The multigenerational experiments were conducted using the plant *Spirodela polyrrhiza* (a duckweed) and the herbivore *Lymnaea stagnalis* (a snail).

Previous studies have shown that selective pressures caused by herbivory can drive the evolution of plant defence in the form of increased presence of secondary metabolites. Likewise, previous studies have demonstrated that plants often engage in inducible defences, a type of phenotypic plasticity wherein plants upregulate defences after experiencing herbivory. The first main contribution of the current manuscript is to demonstrate – compellingly, in my opinion – that both inducible defences and changes in genotype frequencies contribute to rapid changes in plants resistance to herbivory. The experiments used to determine this were thoughtfully designed, and included the appropriate use of controls. The genetic approaches (e.g., used to quantify genotype frequencies) included clever calibrations to reduce bias. The use of statistical approaches (mixed models, mainly) was appropriate. Overall, I think the careful and methodical disentangling of the roles of evolution and phenotypic plasticity is a very important contribution that will generate major interest among those studying induced defences (and phenotypic plasticity in general), plant-herbivore interactions, and experimental evolution.

The paper also makes a second major contribution, which is to demonstrate that herbivory can alter the community composition of microbial communities associated with the plant, which can have further effects on resistance to future herbivory. I think this is also an important result that will have a major impact in our understanding of herbivory and will open up additional avenues of future research.

I would also add that the paper is very well written, logically organized, and clear.

My only suggestions are extremely minor:

~ Line 16 reads awkwardly. It could use one less use of "rapidly"/"rapid" in my opinion.

~ Line 133. I think "timepoint" and "week" mean the same thing here, and should therefore have the same name. This may have cascading effects elsewhere in the paper (e.g., in the caption of Figure 1, I think the same concept is referred to by a third name, "time").

~ Line 167. I have the same type of suggestion as in my previous comment, but this time with the related terms "round" and "bioassay round". There should be a one-to-one mapping of concepts and terms.

Reviewer #4 (Remarks to the Author):

This study demonstrated that snail herbivory can select for phenotypic and genotypic changes in plants. These changes were associated with increases in resistance to subsequent herbivory.

The strength of this study was showing experimentally that herbivory has the potential to cause genotypic changes (evolution). Selection experiments are extremely difficult to conduct and elegant if they can be accomplished. This study combined work in outdoor arenas with detailed analyses of plant traits.

The study has several limitations that are inherent to almost all selection experiments. Only four genotypes were considered which limits how broadly the authors can extrapolate their results. They also used an asexually reproducing species allowing them to process numerous generations but again limiting the potential generality of the results. However, due to the logistical difficulties involved with conducting selection experiments and the novelty of the work, these limitations seem minor and forgivable.

A shortcoming that I found more concerning is the implication of a causal connection between the phenotypic and genotypic traits that were shown to be affected and resistance to herbivory. The evidence that the metabolites or microbiota were causally responsible to resistance was weak. It is equally plausible that the measured traits (smaller, slower growing, heavier) did not cause greater resistance but were merely correlated with the actual resistance mechanisms. The same can be said of the connection between tyramine and resistance. Because of the weakness of this relationship, I found these sections to be a distraction from the main message of the paper. The actual wording of the discussion did not oversell the inferences that can be drawn (see for instance the paragraph beginning on L 439). However, the abstract (L 27-29) implies that induced responses and changes to the microbiome caused increases in resistance and that the microbiota mediate selection (L470). I was not convinced of these hypotheses.

The results were presented very clearly, particularly considering the complexity of the study.

The discussion was rather brief. You could explain that this is some of the best evidence that induced responses can be the subject of selection and can reduce herbivory. I would suggest expanding this at the expense of the current emphasis on metabolomics and the microbiota. I would also have liked a discussion of the fact that selection in this system is unusual since the multi-generational change that you measured involves asexual reproduction. This unusual life history allowed you to carry out this experiment but may also limit the generality of your results. I view this not as a criticism of the study but rather a way to fairly place your results in a more general context.

In general, my assessment of this study is far more positive than my comments sound. Selection experiments are difficult and this one found clear and convincing results that should be of interest to many researchers.

Reviewer #1

Comment #1. This manuscript presents potentially interesting work investigating the evolution of herbivory-induced changes in plant morphology, growth, and microbial communities. The authors performed an impressive array of experiments and targeted bioassays.

Response. We appreciate your positive comments.

Comment #2. However, the methods and results were generally difficult to follow, which made assessment of novelty and interpretation of results confusing. Both sections (methods and results) need major revisions that focus on developing a clear narrative structure to describe the experiments performed and their purpose, which then needs to be reflected in the structure of the results (i.e. structure of the results should clearly link to methods, making it easier for readers to understand). For example, the results section starts off with what appears to be the additional growth bioassays performed (starting week 8?) rather than data collected over both seasons 1 & 2 in the outdoor experiment, which is confusing. It is helpful if results are presented in the same order in which they are described in the methods. In addition, there was misalignment between results reported and statistical methods, including many missing details (e.g. Section “Snail herbivory drives rapid adaptation in *S. polyrhiza* outdoors”). Clear connections to the research aims and questions presented in the introduction are also generally lacking. Below are my specific suggestions for improving the methods and results:

Response. Thanks for the constructive suggestions. We now revised the whole manuscript based on the comments from all the four reviewers. See more detailed responses and changes below. We hope that our work is now more readable and suitable for publication.

Comment #3. Lines 91-96: Description of measurement of indirect effects of herbivory using floating boxes (snails excluded) was confusing – why was this included in the design and why for weeks 4 & 8 only? These data don’t seem to be relevant – the authors need to justify the purpose and how this approach addresses the broader research aims presented in the introduction.

Response. Thanks for pointing this out. The floating boxes were used to test the indirect effect of herbivory on individual genotypes *in situ*. As plants growing inside the boxes are directly exposed to changes in microbiota and nutrients, such experiments are pivotal for testing our main hypotheses. We now clarified this in the method section. See line 319.

Comment #4. Subsection starting on Line 135 – “Growth and herbivory assays outdoors”. Were growth bioassays performed in absence of herbivory? It is confusing that herbivory assays were performed “right before” growth assays, does this mean plants were exposed to herbivory before growth assays? Clarification is needed.

Response. The growth bioassay was performed in absence of herbivory. The bioassays for growth and herbivory were performed using different plant individuals that were taken from the same population. Once herbivory assays on individual genotypes were completed, we were then able to use the same floating boxes to run bioassays on mixed genotypes. We started by collecting fronds from the mixed population and running growth assays, followed by herbivory assays on the same fronds. We also clarify that growth assays were performed in absence of herbivores as floating boxes were screened to prevent snails entering and feeding on plants.

Comment #5. Growth rates do not appear to be measured correctly – authors indicate relative changes of surface area and biomass were estimated but it unclear how that relates to growth rate (e.g. lines 144-146). Changes in biomass and surfaces areas from beginning and end of 10 day bioassay should be used to estimate growth, or simply analyze measures taken at day 10. Clarification is needed.

Response. Thanks for pointing this out. We agree that the text was unclear. We now clarify that the effect of herbivory was tested separately for relative growth rate, and relative changes in plant biomass and surface area (see results in Supplementary Material).

Comment #6. Resistance was measured as # fronds consumed – unclear if this is the entire frond or if partial consumption of leaf tissue was measured (which could effect surface area measures). It would be helpful to explain patterns of snail herbivory, do they consume portions of fronds or entire fronds? This information matters for interpretation of differences in surface area (i.e. won't plants exposed to direct herbivory have reduced surface area?).

Response. We fully aware the issue that herbivory also reduces surface area. Therefore, we only considered fronds that were undamaged for monitoring the phenotypic changes. We now clarified this in the method section. See line 348.

Comment #7. Metabolomics section (line 228-253) lacked critical information on methods used – what type of metabolomics was performed, where, on what machine? What quality control measures were used to filter data? How was the concentration or abundance of metabolites estimated? All of these details need to be included.

Response. Thanks for this comment. Considering that the metabolomics methods (machine, quality control etc.) are quite extensive and might interrupt the information flow, we have moved them to the Supplementary Material, so more curious readers can easily access this information. We now provide a summary in the methods section. See line 444.

Comment #8. Unclear why microbial metabarcoding was performed on just 2 genotypes (section starting line 237) – please explain or justify. It is also generally unclear why shotgun sequencing was used on mixed populations only, and then additional metabarcoding was performed on a subset of 2 genotypes. Authors need to provide explanation and purpose of these measurements, what questions are being addressed?

Response. We used shotgun metagenomics to infer microbial communities from mixed populations, because this data was already generated from pool-seq. Indeed, the data not aligning the duckweed reference genome was used to characterize the duckweed-associated microbial communities. When dealing with individual genotypes, our question was very specific (test differences in the plants microbiome between the two genotypes exposed to herbivory or not), and considering we were dealing with individual genotypes. Therefore, amplicon metagenomics is more cost-effective than shotgun sequencing. We now clarify this in the method section. See lines 429 and 463.

Comment #9. It is unclear what novel information the indoor bioassays provided, especially because populations were artificially created to replicate genotype frequencies observed at week 12 in the outdoor experiments. The authors need to

provide additional details clarifying the purpose of these experiments/questions addressed.

Response. We performed the indoors bioassays to disentangle the effects of phenotypic plasticity and changes in genotype frequency on the observed resistance changes. From the outdoors experimental evolution we were not able to disentangle the effect driven by herbivory-induced responses (phenotypic plasticity) from the effect driven by changes in genotype frequency (evolution). In the indoors assays, we worked with individual genotypes that either experienced herbivory or not, thus generating two groups where herbivory-driven phenotypic plasticity was induced or not. Then, each of these groups of individual genotypes were used to assemble plant populations whose genotype frequency looked either like the one observed in outdoors under control conditions, or the one observed herbivore attack. In this way, by having these four populations (Fig. 4) we were able to disentangle the effects of phenotypic plasticity and evolution on our observations, which would be impossible to test outdoors because of logistics. More importantly, despite of significant growth conditions, we observed the similar herbivory-induced phenotypic changes (e.g., smaller fronds) both indoors and outdoors. We now clarified this in the discussion section. See line 488.

Comment #10. Section “Snail herbivory drives rapid adaptation in *S. polyrhiza* outdoors” – this section was generally missing details from statistical analyses described in the methods, such as linear mixed models and metabolomics. For example, chi-squared values were reported from analyses that were not described in the methods – results should align with what is described in methods. Additionally, the metabolomics results consisted of a single sentence with no statistics included (lines 324-326) – the authors need to expand on these results, what other metabolites were different?

Response. Data analysis are reported in the method section of “metabolomics. However, to make the manuscript more concise, we only provided a brief information on metabolomics and moved the details in the supplementary material.

Comment #11. Figure 1 A & B was confusing - are these results from additional bioassays (described starting line 135) or data collected across seasons (lines 112-123)? Please clarify – if these data are not from the additional bioassays in the absence of herbivory pressure, then they potentially do not support claims of rapid evolution (i.e. biomass expected to be less from fronds directly exposed to herbivory). Also, chi-squared values are reported which do not align with linear mixed models described in methods. Please provide results for fixed and random effects of the linear mixed models.

Response. The data were collected from the ponds. However, to avoid the confounding factor that herbivory also reduces biomass and surface area, we measured these values by collecting 500 **undamaged** individuals. This is clarified in the figure legend and method section. We also clarified the statistics tests.

Comment #12. Lines 371-373 – where is this indoor experiment described in the methods and how is it different from that described on line 356-368? Authors needs to clarify, and again the methods section should also provide explanation of purpose of indoor experiments that then clearly align with results.

Response. The experiment at L356-368 was performed using fronds from our sterile collection to test intrinsic differences among genotypes, while the other indoor

experiment was performed in the greenhouse to disentangle the effects driven by phenotypic plasticity from the one driven by changes in genotype frequency on our observations. We now added this missing information in the Supplementary Material (Fig. S4).

Comment #13. Microbiome data was not well integrated into the manuscript overall - not only were there missing details but the authors make incorrect claims that herbivory-induced microbiota changes contributed to increased resistance to herbivory. Below are my specific suggestions. Details on differences in microbial communities and results from statistical analyses need to be included (e.g. expand on sentence on lines 393-94). What microbial taxa were identified and which are differentially abundant between control plants and those exposed to herbivory? What are the results from PERMANOVA & DeSeq? The authors should include a more complete analysis of microbial diversity and composition, including both alpha (e.g. species richness) and beta (e.g. Bray Curtis, Unifrac) diversity measures.

Response. The results from PERMANOVA (Tab. S5) and DESeq2 (Tab. S8) are available as supplementary material. Tests on beta diversity were indeed ran using PERMANOVA (e.g., Tab. S5) and multivariate visualizations (e.g., Fig. 5) calculated on a Unifrac distance matrix, as specified in the methods section. We agree that testing the influence of herbivory on the (alpha)diversity of the microbial communities could be interesting. However, this is out of the scope of this paper, as it does not contribute to address our key questions.

Comment #14. Lines 395- 413 – based on the experiments described it is not possible to draw conclusions about the effect of the plant microbiome on resistance to herbivory because proper controls (i.e. sterile water/reduced microbiota treatment) were not included. The authors use language that suggest causation (i.e. changes in microbes caused changes in resistance) but experiments only show associations. Fig 5 shows herbivory induced changes in the plant microbiome, but this overall effect does not seem to vary substantially between genotypes expressing different levels of resistance (marginally significant effect of herbivory in Fig 5b). The authors provide no evidence to support claims that microbiota are involved in resistance to herbivory, to do that follow-up experiments under sterile (no microbiome) conditions or transfer of microbiomes between genotypes are needed. This section needs to be substantially edited (i) to remove language suggesting a causal relationship between herbivory-induced changes in the plant microbiome and resistance and (ii) to include additional details on what herbivory-induced microbial differences were observed (e.g. results from DeSeq, additional analysis for both alpha (e.g. species richness) and beta (e.g. Bray Curtis, Unifrac) diversity measures).

Response. Thanks for your comment. Following your suggestion, we toned down our claims in the abstract and discussion. See line 245. However, we believe that sterile water / reduced microbiota treatment is not required to quantify the effects of **microbiota changes** on resistance to herbivory. This is because we directly grew the plants in two environment that differ most likely only in microbiota (as nutrients are the same).

Reviewer #2

Comment #1. I understand the overall objective of this study but I do not really see how the experimental approaches chosen allow to meet the specific objectives. For instance, what is the purpose of the indoor system in comparison to the outdoor

system? In addition, the results are not sufficiently described and discussed. To my point of view, it seems that elements required to fully describe the experimental system are missing.

Response. Thanks for this feedback. We now thoroughly revised the manuscript based on the comments from all the four reviewers.

Comment #2. What is the composition of the Com BIO liquid fertilizer?

Response. We included in the text the information available in the product label.

Comment #3. It's not clear to me what is the volume of liquid inside the ponds? I understand there is 40 L of soil and 0.55 L of liquid fertilizer but nothing else?

Response. Ponds were already filled with water from the previous season. We now clarify this in the manuscript.

Comment #4. How floating boxes float on without water?

Response. The floating boxes were framed with Styrofoam, so they were floating on the water surface. Each box had nets at the bottom allowing water and phytoplankton community to go through, while preventing herbivores to directly access plants. We now clarified this in the method section. See line 319.

Comment #5. Is the fronds density the same between the floating boxes and the rest of the cage at the beginning of the experiment and over time?

Response. The initial number of duckweed individuals was a bit lower in the floating boxes compared to the wider cage and we did not specifically control their density over time. However, the duckweed in both floating boxes and cages quickly covered the complete surface. When samples were collected for bioassays (week 8), all floating boxes and all cages has similar duckweed density.

Comment #6. This analysis was carried out on a very limited number of time samples: after 8 and 12 weeks (metagenomes) and only 8 weeks (amplicon metagenomics). Why 2 types of analyses/methods were used? Why the amplicon metagenomics analyses was only shown for 2 genotypes, Sp21 and Sp65? Why the microbial community structure was not analysed earlier like at time 0 and also during year 2 at the end of the experimental evolution in order to have a better description and understanding of population dynamics of the associated microbial communities during the whole experimental evolution, and in response to herbivory.

Response. We agree that a higher number of data points might have helped us in better describing the dynamics within the plant microbiome. However, these analyses are costly. Therefore, we decided to focus on those timepoints that would enable us to test specific questions related to the effects of the plant microbiome on plant resistance towards herbivores. We used shotgun metagenomics to infer microbial communities from mixed populations, because this data was already generated from pool-seq. Indeed, the data not aligning the duckweed reference genome was used to characterize the duckweed-associated microbial communities. When dealing with individual genotypes, our question was more specific (test differences in the plants microbiome between genotypes exposed or not to herbivory), and considering we were dealing with individual genotypes, pool-seq was not necessary to address our question. Therefore, we opted for a more cost-effective amplicon metagenomics. We focused on those two genotypes because their frequency was influenced by

multigenerational herbivory, while the other two genotypes were unaffected. Therefore, these genotypes could have potentially affected the microbial community of the mixed population in a genotype-frequency dependent manner. We now clarified this in the manuscript. See lines 429 and 463.

Comment #7. In the introduction section (line 58), the authors indicate that microbiome community are highly dynamic and sensitive to abiotic environment. It's true but the authors only performed microbiome analysis on a reduced number of sampling times and of conditions. Why do they choose these sampling times and those conditions?

Response. We selected those time points to integrate the bioassay and microbiome data. As our goal is not to describe the dynamics of herbivory-induced changes in plant microbiota, we did not sequence samples from other time points.

Comment #8. Authors show that there are changes in the frequency of *Spirodela polyrhiza* populations in response to herbivory. These changes may lead to direct changes in microbial communities as well as phenotypic changes in *S. polyrhiza* in response to herbivory. How can the authors differentiate the effect of these 2 confounding factors on the structure of the microbial communities? It would have been interesting to compare the structure of microbial community of each genotype (not only on Sp21 and Sp65) and of mix populations at different times in presence and absence of herbivory pressure.

Response. From our pool-seq data we derived a dataset that allowed us to test changes in the microbiome of mixed populations as effect of herbivory. When focusing on individual genotypes, we selected Sp21 and Sp65 because they were those showing differences in genotype frequency as effect of herbivory. We agree that having a larger panel of genotypes would be interesting to test how herbivory shapes microbial communities and how it changes when plant genotypes vary. However, as this is not the focus of our manuscript, we specifically focused on the two genotypes to reduce costs.

Comment #9. To my point of view, it is necessary to describe the experimental procedure used for the metabolomics analysis inside materials and methods section of the paper instead of the supplementary materials.

Response. The metabolomics method description is very extensive. We thought it might be distracting for non-specialist readers. Therefore, we prefer to keep the specific details in the Supplementary Material. However, in the revised version, we now provided more information in the main text so that the readers can have a better understanding of the method we used. See line 444.

Comment #10. In addition, the results from Fig S3 are not sufficiently described and discussed. What does this study of the evolution of metabolomes after 12 and 58 weeks show regarding plant responses to herbivory? It appears that after 12 weeks more defensive metabolites are identified as overproduced in the conditions with herbivory vs without than after 58 weeks? Tyramine is the sole compound that is maintained increased after 12 and 58 weeks ! What does this mean? Could you discuss these results in more depth in the discussion section ? Are the metabolome changes correlated to the evolution of *Spirodela polyrhiza* populations inside the mix throughout the experimental evolution?

Response. We agree, answering those questions would be very interesting. However, for most of metabolites, the changes are highly dynamic, likely due to environmental changes under outdoor conditions. Additionally, for many metabolites, we have little evidence on their role under snail herbivory. We added a few more comments on this to the discussion. Despite of the uncertainties in the interpretation we further provide the data set for readers as it might provide useful information for future studies.

Comment #11. In the introduction (lines 41 to 43), authors indicate that plants often respond to herbivores by increasing the biosynthesis of defensive metabolites. What are these defensive metabolites? Could you detail the main compound family of these defensive metabolites for aquatic plants like *Spirodela polyrhiza*? Are these compounds identified in the metabolome analysis performed in the presented work?

Response. We agree, knowing defensive metabolites in *S. polyrhiza* would be very important. Unfortunately, as only a few groups are working on plant-herbivore interactions using this plant as a model, we currently know little about this. The main specialized metabolites in *S. polyrhiza* are flavonoids. We have quantified them in this study (Table S11). However, we do not have evidence showing their defensive function against snail herbivory. We are currently working on developing genetic transformation tools to address this challenge. We now provide more discussions in the revised manuscript. See line 229.

Comment #12. In the discussion section (lines 445 to 446), authors indicate that ‘using a targeted metabolomics approach, we observed changes in several candidate defensive metabolites, which were induced by herbivory across two growing seasons.’ What are those compounds?

Response. Thanks for pointing this out. We now list the candidate defensive compounds in the discussion, along with references to papers that suggest their role in plant defence against herbivores.

Comment #13. They also indicate that ‘their results support the importance of cryptic genetic variation in contributing rapid adaptive evolution to environmental changes’ but why do the authors introduce this notion of ‘cryptic genetic variation’ there? Against what does this mean? Could they discuss further this point.

Response. We think this was due to unclear wording. “Cryptic genetic variation” is a key concept that explains how selection could act on phenotypic plasticity. For example, if a genetic change will only cause a phenotype under stressful conditions, it will only be selected under stress condition but not under normal condition. Our observed data is consistent with such concept. We now clarified this in our discussion.

Comment #14. Lines 303-304: authors indicate that ‘Nutrient levels and pH were similar in both cages ... suggesting that the effects we observed were not driven by differences in resources’. Overall, the NMDS analysis (FigS6) shows no difference. However, in Table S9 it seems that the level of Cl and SO₄ in water were changed in response to herbivory. Could you add a comment on these changes in the result and discussion sessions.

Response. Thanks for pointing this out. We implemented Tab S9 with mean \pm sd values for each nutrient and for both control and herbivory cages. We also clarified in the manuscript that although Cl and SO₄ showed differences between cages, the effect size was so small that those changes unlikely contributed to the biological effects we observed on plant populations. See line 84.

Reviewer #3

Comment #1. This study investigated the role of evolution and phenotypic plasticity in driving rapid changes in plant resistance to herbivory. The multigenerational experiments were conducted using the plant *Spirodela polyrhiza* (a duckweed) and the herbivore *Lymnaea stagnalis* (a snail). Previous studies have shown that selective pressures caused by herbivory can drive the evolution of plant defence in the form of increased presence of secondary metabolites. Likewise, previous studies have demonstrated that plants often engage in inducible defences, a type of phenotypic plasticity wherein plants upregulate defences after experiencing herbivory. The first main contribution of the current manuscript is to demonstrate – compellingly, in my opinion – that both inducible defences and changes in genotype frequencies contribute to rapid changes in plants resistance to herbivory. The experiments used to determine this were thoughtfully designed, and included the appropriate use of controls. The genetic approaches (e.g., used to quantify genotype frequencies) included clever calibrations to reduce bias. The use of statistical approaches (mixed models, mainly) was appropriate. Overall, I think the careful and methodical disentangling of the roles of evolution and phenotypic plasticity is a very important contribution that will generate major interest among those studying induced defences (and phenotypic plasticity in general), plant-herbivore interactions, and experimental evolution. The paper also makes a second major contribution, which is to demonstrate that herbivory can alter the community composition of microbial communities associated with the plant, which can have further effects on resistance to future herbivory. I think this is also an important result that will have a major impact in our understanding of herbivory and will open up additional avenues of future research. I would also add that the paper is very well written, logically organized, and clear.

Response. We thank you for your comments and positive feedback on our study.

Comment #2. Line 16 reads awkwardly. It could use one less use of "rapidly"/"rapid" in my opinion.

Response. We rephrased it as suggested.

Comment #3. Line 133. I think "timepoint" and "week" mean the same thing here, and should therefore have the same name. This may have cascading effects elsewhere in the paper (e.g., in the caption of Figure 1, I think the same concept is referred to by a third name, "time").

Response. We rephrased it as suggested.

Comment #4. Line 167. I have the same type of suggestion as in my previous comment, but this time with the related terms "round" and "bioassay round". There should be a one-to-one mapping of concepts and terms.

Response. We rephrased it as suggested.

Reviewer #4

Comment #1. This study demonstrated that snail herbivory can select for phenotypic and genotypic changes in plants. These changes were associated with increases in resistance to subsequent herbivory. The strength of this study was showing experimentally that herbivory has the potential to cause genotypic changes (evolution). Selection experiments are extremely difficult to conduct and elegant if they can be accomplished. This study combined work in outdoor arenas with detailed analyses of

plant traits. The study has several limitations that are inherent to almost all selection experiments. Only four genotypes were considered which limits how broadly the authors can extrapolate their results. They also used an asexually reproducing species allowing them to process numerous generations but again limiting the potential generality of the results. However, due to the logistical difficulties involved with conducting selection experiments and the novelty of the work, these limitations seem minor and forgivable.

Response. Thank you for the positive feedback.

Comment #2. A shortcoming that I found more concerning is the implication of a causal connection between the phenotypic and genotypic traits that were shown to be affected and resistance to herbivory. The evidence that the metabolites or microbiota were causally responsible to resistance was weak. It is equally plausible that the measured traits (smaller, slower growing, heavier) did not cause greater resistance but were merely correlated with the actual resistance mechanisms. The same can be said of the connection between tyramine and resistance. Because of the weakness of this relationship, I found these sections to be a distraction from the main message of the paper. The actual wording of the discussion did not oversell the inferences that can be drawn (see for instance the paragraph beginning on L 439). However, the abstract (L 27-29) implies that induced responses and changes to the microbiome caused increases in resistance and that the microbiota mediate selection (L470). I was not convinced of these hypotheses.

Response. As suggested, we rephrased the abstract.

Comment #3. The results were presented very clearly, particularly considering the complexity of the study.

Response. Thanks.

Comment #4. The discussion was rather brief. You could explain that this is some of the best evidence that induced responses can be the subject of selection and can reduce herbivory. I would suggest expanding this at the expense of the current emphasis on metabolomics and the microbiota. I would also have liked a discussion of the fact that selection in this system is unusual since the multi-generational change that you measured involves asexual reproduction. This unusual life history allowed you to carry out this experiment but may also limit the generality of your results. I view this not as a criticism of the study but rather a way to fairly place your results in a more general context.

Response. We implemented the discussion to include this aspect. See line 256.

Comment #5. In general, my assessment of this study is far more positive than my comments sound. Selection experiments are difficult and this one found clear and convincing results that should be of interest to many researchers.

Response. Thanks.

REVIEWERS' COMMENTS:

Reviewer #1 (Remarks to the Author):

No further comments, authors have addressed most suggestions/edits adequately.

Reviewer #2 (Remarks to the Author):

The manuscript has been in-depth modified. My comments have been taken into account and I obtain the response to my queries.